# INFORMATION DROPOUT: LEARNING OPTIMAL REPRESENTATIONS THROUGH NOISE*

**Alessandro Achille & Stefano Soatto**
Department of Computer Science
University of California, Los Angeles
{achille,soatto}@cs.ucla.edu

## ABSTRACT

We introduce Information Dropout, a generalization of dropout that is motivated by the Information Bottleneck principle and highlights the way in which injecting noise in the activations can help in learning optimal representations of the data. Information Dropout is rooted in information theoretic principles, it includes as special cases several existing dropout methods, like Gaussian Dropout and Variational Dropout, and, unlike classical dropout, it can learn and build representations that are invariant to nuisances of the data, like occlusions and clutter. When the task is the reconstruction of the input, we show that the information dropout method yields a variational autoencoder as a special case, thus providing a link between representation learning, information theory and variational inference. Our experiments validate the theoretical intuitions behind our method, and we find that information dropout achieves a comparable or better generalization performance than binary dropout, especially on smaller models, since it can automatically adapt the noise to the structure of the network, as well as to the test sample.

## 1 INTRODUCTION

In the general supervised setting, we want to learn the conditional distribution $p(\mathbf{y}|\mathbf{x})$ of some random variable $\mathbf{y}$, which we refer to as the *task*, given (samples of the) input data $\mathbf{x}$. In typical applications, $\mathbf{x}$ is often high dimensional (for example an image or a video), while $\mathbf{y}$ is low dimensional, such as a label or a coarsely-quantized location. In such cases, a large part of the variability in $\mathbf{x}$ is actually due to *nuisance factors* that affect the data, but are otherwise irrelevant for the task. Since by definition these nuisance factors are not predictive of the task, they should be disregarded during the inference process. However, it often happens that modern machine learning algorithms, in part due to their high flexibility, will fit spurious correlations, present in the training data, between the nuisances and the task, thus leading to poor generalization performance.

In view of this, Tishby & Zaslavsky (2015) argue that the success of deep learning is in part due to the capability of neural networks to build incrementally better representations that expose the relevant information, while at the same time discarding nuisance variability. This interpretation is intriguing, as it establishes a connection between machine learning, probabilistic inference, and information theory. However, the commonly used training methods do not seem to stem from this insight, and indeed deep networks may maintain even in the topmost layers dependencies on easily ignorable nuisances (see for example Figure 2).

To bring the practice in line with the theory, and to better understand these connections, we introduce a new layer, that we call *Information Dropout*, which allows the network to selectively introduce multiplicative noise in the layer activations, and thus to control the flow of information. We then introduce a modified cost function, that can be seen as an approximation of the Information Bottleneck Lagrangian of Tishby et al. (1999), which encourages the creation of representations of the data which are increasingly insensitive to the action of nuisances. In practice, this prevents the network from overfitting to the nuisances. As we show in various experiments, our method improves the

---

*Dedicated to Naftali Tishby in the occasion of the conference *Information, Control and Learning* held in his honor in Jerusalem, September 26-28, 2016.

generalization performance, and is comparable or better than the dropout baseline. In particular, it considerably improves over binary dropout on smaller models, since, unlike dropout, Information Dropout also adapts the noise to the structure of the network and to the individual sample at test time.

Apart from the practical interest of Information Dropout, one of our main results is that Information Dropout can be seen as a generalization to several existing dropout methods, providing a unified framework to analyze them, together with some additional insights on empirical results. As we discuss in Section 2, the introduction of noise to prevent overfitting has already been studied from several points of view. For example the original formulation of dropout of Srivastava et al. (2014), which introduces binary multiplicative noise, was motivated as a way of efficiently training an ensemble of exponentially many networks, that would be averaged at testing time. Kingma et al. (2015) introduce *Variational Dropout*, a dropout method which closely resemble ours, and is instead derived from a Bayesian analysis of neural networks. Information Dropout gives an alternative information-theoretic interpretation of those methods.

As we show in Section 5, other than being very closely related to Variational Dropout, Information Dropout directly yields a variational autoencoder as a special case when the task is the reconstruction of the input. This result is in part expected, since by construction Information Dropout seeks an optimal representation of the input for the task of reconstruction, and the representation given by the latent variables of a variational autoencoder fits the criteria. However, it still rises the question of exactly what and how deep are the links between information theory, representation learning, variational inference and nuisance invariance. This work can be seen as a small step in answering this question.

## 2 Related work

The main contribution of our work is to establish how two seemingly different areas of research, namely the study of noise and dropout methods to prevent overfitting, and the study of optimal representations, can be linked through the Information Bottleneck principle.

Dropout was introduced by Srivastava et al. (2014). The original motivation was that by randomly dropping the activations during training, we can effectively train an ensemble of exponentially many networks, that are then averaged during testing, therefore reducing overfitting. Wang & Manning (2013) suggested that dropout could be seen as performing a Monte-Carlo approximation of an implicit loss function, and suggest that instead of multiplying the activations by binary noise, like in the original dropout, multiplicative Gaussian noise with mean 1 can be used as a way of better approximating the implicit loss function. This leads to a comparable performance but faster training than binary dropout.

Kingma et al. (2015) take a similar view of dropout as introducing multiplicative (Gaussian) noise, but instead study the problem from a Bayesian point of view. Given a training dataset $\mathcal{D} = \{(\mathbf{x}_i, \mathbf{y}_i)\}_{i=1,\ldots,N}$ and a prior distribution $p(\mathbf{w})$, we want to compute the posterior distribution $p(\mathbf{w}|\mathcal{D})$ of the weights $\mathbf{w}$ of the network. As is customary in variational inference, the true posterior can be approximated by minimizing the negative variational lower bound $\mathcal{L}(\theta)$ of the marginal log-likelihood of the data,

$$\mathcal{L}(\theta) = \frac{1}{N} \sum_{i=1}^{N} \mathbb{E}_{\mathbf{w} \sim p_\theta(\mathbf{w}|\mathcal{D})}[-\log p(\mathbf{y}_i|\mathbf{x}_i, \mathbf{w})] + \frac{1}{N} \mathrm{KL}(p_\theta(\mathbf{w}|\mathcal{D}) \parallel p(\mathbf{w})).$$

This minimization is difficult to perform, since it requires to repeatedly sample new weights for each sample of the dataset. As an alternative, Kingma et al. (2015) suggest that the uncertainty about the weights that is expressed by the posterior distribution $p_\theta(\mathbf{w}|\mathcal{D})$ can equivalently be encoded as a multiplicative noise in the activations of the layers (the so called *local reparametrization trick*). As we will see in the following sections, this loss function closely resemble the one of Information Dropout, which however is derived from a purely information theoretic argument based on the Information Bottleneck principle. One difference is that we allow the parameters of the noise to change on a per-sample basis (which, as we show in the experiments, can be useful to deal with nuisances), and that we allow a scaling constant $\beta$ in front of the KL-divergence term, which can be changed freely. Interestingly, even if the Bayesian derivation does not allow a rescaling of the KL-divergence, Kingma et al. (2015) notice that choosing a different scale for the KL-divergence term can indeed lead to improvements in practice.

The interpretation of deep neural network as a way of creating successively better representations of the data has already been suggested and explored by many, as described in Tishby & Zaslavsky (2015). Some have focused on creating representations that are *maximally* invariant to nuisances, especially when they have the structure of a (possibly infinite-dimensional) group acting on the data, like Sundaramoorthi et al. (2009), or, when the nuisance is a locally compact group acting on each layer, by successive approximations implemented by hierarchical convolutional architectures, like Anselmi et al. (2016) and Bruna & Mallat (2011). In these cases, which cover common nuisances such as translations and rotations of an image (affine group), or small diffeomorphic deformations due to a slight change of point of view (group of diffeomorphisms), the representation is equivalent to the data modulo the action of the group. However, when the nuisances are not a group, as is the case for occlusions, it is not possible to achieve such equivalence, that is, there is a loss. To address this problem, Soatto & Chiuso (2016) defined optimal representations not in terms of maximality, but in terms of *sufficiency*, and characterized representations that are both sufficient and invariant. They argue that the management of nuisance factors common in visual data, such as change of viewpoint, local deformations, and changes of illumination, is directly tied to the specific structure of deep convolutional networks, where local marginalization of simple nuisances at each layer results in marginalization of complex nuisances in the network as a whole.

Our work fits in this last line of thinking, where the goal is not equivalence to the data up to the action of (group) nuisances, but instead sufficiency for the task. **Our main contribution** in this sense is to show that injecting noise into the layers, and therefore using a non-deterministic function of the data, can actually simplify the theoretical analysis and lead to improved invariance to nuisances. This is an alternate explanation to that put forth by the references above.

## 3 Optimal representations and the Information Bottleneck loss

Given some input data $\mathbf{x}$, we want to compute some (possibly nondeterministic) function of $\mathbf{x}$, called a *representation*, that has some desirable properties in view of the task $\mathbf{y}$, for instance by being more convenient to work with, exposing relevant statistics, or being easier to store. Ideally, we want this representation to be as good as the original data for the task, and not squander resources modeling parts of the data that are irrelevant to the task. Formally, this means that we want to find a random variable $\mathbf{z}$ satisfying the following conditions:

(i) $\mathbf{z}$ is a **representation** of $\mathbf{x}$; that is, its distribution depends only on $\mathbf{x}$, as expressed by the following Markov chain:

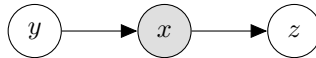

(ii) $\mathbf{z}$ is **sufficient** for the task $\mathbf{y}$, that is $I(\mathbf{x}; \mathbf{y}) = I(\mathbf{z}; \mathbf{y})$, expressed by the Markov chain:

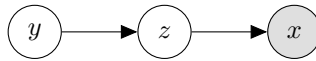

(iii) among all random variables satisfying these requirements, the mutual information $I(\mathbf{x}; \mathbf{z})$ is **minimal**. This means that $\mathbf{z}$ discards all the variability that is not relevant to the task.

Using the identity $I(\mathbf{x}; \mathbf{y}) - I(\mathbf{z}; \mathbf{y}) = H(\mathbf{y}|\mathbf{z}) - H(\mathbf{y}|\mathbf{x})$, where $H$ denotes the entropy and $I$ the mutual information, it is easy to see that the above conditions are equivalent to finding a distribution $p(\mathbf{z}|\mathbf{x})$ which solves the optimization problem

$$\text{minimize} \quad I(\mathbf{x}; \mathbf{z})$$
$$\text{s.t.} \quad H(\mathbf{y}|\mathbf{z}) = H(\mathbf{y}|\mathbf{x}).$$

Since the above minimization is difficult to perform in general due to the non-linear constraint, Tishby et al. (1999) propose to minimize instead the following relaxation, known as the *Information Bottleneck Lagrangian*,

$$\mathcal{L} = H(\mathbf{y}|\mathbf{z}) + \beta I(\mathbf{x}; \mathbf{z}). \tag{1}$$

where $\beta$ is some positive constant. It is easy to see that, in the limit $\beta \to 0^+$, this is equivalent to the original problem. When all random variables are discrete and $\mathbf{z} = T(\mathbf{x})$ is a deterministic function of $\mathbf{x}$, the algorithm proposed by Tishby et al. (1999) can be used to minimize the IB Lagrangian

efficiently. However, no algorithm is known to minimize the IB Lagrangian for non-Gaussian, high-dimensional continuous random variables.

**Our main result** is that, when we restrict to the family of distributions obtained by injecting noise to one layer of a neural network, we can efficiently approximate and minimize the IB Lagrangian. Since we restrict the family of distributions, this does not in general guarantee that the resulting representation will be optimal. We can however iterate the process to obtain incrementally improved representations. As we will show, this process can be effectively implemented through a generalization of the dropout layer that we call *Information Dropout*.

In preparation for this, we rewrite the IB Lagrangian as a per-sample loss function. Let $p(\mathbf{x}, \mathbf{y})$ denote the true distribution of the data, from which the training set $\{(\mathbf{x}_i, \mathbf{y}_i)\}_{i=1,\ldots,N}$ is sampled, and let $p_\theta(\mathbf{z}|\mathbf{x})$ and $p_\theta(\mathbf{y}|\mathbf{z})$ denote the unknown distributions that we need to estimate, parametrized by $\theta$. Then, we can write the two terms in the IB Lagrangian as

$$H(\mathbf{y}|\mathbf{z}) \simeq \mathbb{E}_{\mathbf{x},\mathbf{y} \sim p(\mathbf{x},\mathbf{y})} \left[ \mathbb{E}_{\mathbf{z} \sim p_\theta(\mathbf{z}|\mathbf{x})} [-\log p_\theta(\mathbf{y}|\mathbf{z})] \right]$$
$$I(\mathbf{x}; \mathbf{z}) = \mathbb{E}_{\mathbf{x} \sim p(\mathbf{x})} [\mathrm{KL}(p_\theta(\mathbf{z}|\mathbf{x}) \parallel p_\theta(\mathbf{z}))],$$

where KL denotes the Kullback-Leibler divergence. We can therefore approximate the IB Lagrangian empirically as

$$\mathcal{L} = \frac{1}{N} \sum_{i=1}^{N} \mathbb{E}_{z \sim p(\mathbf{z}|\mathbf{x}_i)} [-\log p(\mathbf{y}_i|\mathbf{z})] + \beta \, \mathrm{KL}(p_\theta(\mathbf{z}|\mathbf{x}_i) \parallel p_\theta(\mathbf{z})). \tag{2}$$

Notice that the first term simply is the average cross-entropy loss, which is the most commonly used loss function in deep learning. The second term can then be seen as a regularization term that penalizes the transfer of information from $\mathbf{x}$ to $\mathbf{z}$. In the next section, we discuss ways to control such information transfer through the injection of noise.

**Remark.** Aside from being easier to work with, stochastic representations of the data can realize a lower value of the IB Lagrangian than any deterministic representation. For an example, consider the task of reconstructing single random bit $y$ given a noisy observation $x$. The only deterministic representations are equivalent to the either the noisy observation itself or to the trivial constant map. It is not difficult to check that for opportune values of $\beta$ and of the noise, neither realize the optimal tradeoff reached by an opportune stochastic representation.

## 4 INFORMATION DROPOUT

Inspired by the effectiveness of dropout, we propose the following way of constructing a representation $\mathbf{z}$: first, we compute a function $f(\mathbf{x})$ of the input through a sequence of convolutional or fully-connected layers, followed by a nonlinear activation function. Ideally, this should help "disentangle" the nuisances from the (random) function of the data that is relevant to the task, as we illustrate in Appendix C. Subsequently, we selectively let the relevant information flow by applying multiplicative noise from a noise distribution $p_{\alpha(x)}(\varepsilon)$ with mean 1 and whose parameters $\alpha(\mathbf{x}) = \alpha_\theta(\mathbf{x})$, such as the variance, are selected by the network itself:

$$\varepsilon \sim p_{\alpha(\mathbf{x})}(\varepsilon),$$
$$\mathbf{z} = \varepsilon \odot f(\mathbf{x}),$$

where "$\odot$" denotes the element-wise product. Notice that, if $p_{\alpha(\mathbf{x})}(\varepsilon)$ is a Bernoulli distribution rescaled to have mean 1, this reduces exactly to the classic binary dropout layer. As we discussed in Section 2, there are also variants of dropout that use different distributions.

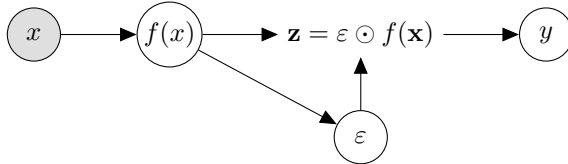

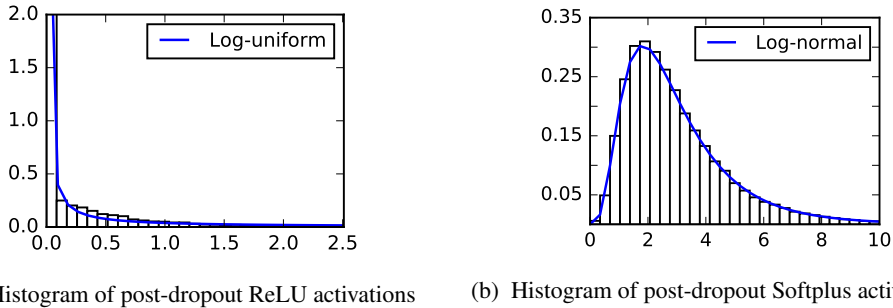

(a) Histogram of post-dropout ReLU activations

(b) Histogram of post-dropout Softplus activations

Figure 1: Comparison of the empirical distribution $p(z)$ of the post-noise activations with our proposed prior when using: (a) ReLU activations, for which we propose a log-uniform prior, and (b) Softplus activations, for which we propose a log-normal prior. In both cases, the empirical distribution approximately follows the proposed prior. Both histograms where obtained from the last dropout layer of the All-CNN-32 network described in Table 2, trained on CIFAR-10.

A natural choice for the multiplicative noise distribution $p_{\alpha(\mathbf{x})}(\varepsilon)$, which also simplifies the theoretical analysis, is the log-normal distribution $p_{\alpha(\mathbf{x})}(\varepsilon) = \log \mathcal{N}(0, \alpha_\theta^2(\mathbf{x}))$. Once we fix this noise distribution, given the above expression for $\mathbf{z}$, we can easily compute the distribution $p_\theta(\mathbf{z}|\mathbf{x})$ that appears in eq. (2). However, to be able to compute the KL-divergence term, we still need to fix a prior distribution $p_\theta(\mathbf{z})$. The choice of this prior largely depends on the expected distribution of the activations $f(\mathbf{x})$. In the following, we assume for simplicity that all the activations are approximately independent and identically distributed. In Appendix B we show that, even if the activations are not independent, optimizing the loss in Equation (2) under the assumption of a factorized prior still makes sense, and it actually encourages the components to become independent. We concentrate on two of the most common activation functions, the *rectified linear unit* (ReLU), which is easy to compute and works well in practice, and the *Softplus* function, which can be seen as a strictly positive and differentiable approximation of ReLU.

A network implemented using only ReLU and a final Softmax layer has the remarkable property of being scale-invariant, meaning that multiplying all weights, biases, and activations by a constant does not change the final result. Therefore, from a theoretical point of view, it would be desirable to use a scale-invariant prior. The only such prior is the improper log-uniform, $p(\log(z)) = c$, or equivalently $p(z) = c/z$, which was also suggested by Kingma et al. (2015), but as a prior for the weights of the network, rather than the activations. Since the ReLU activations are frequently zero, we also assume $p(z = 0) = q$ for some constant $0 \leq q \leq 1$. Therefore, the final prior has the form $p(z) = q\delta_0(z) + c/z$, where $\delta_0$ is the Dirac delta in zero. In Figure 1a, we compare this prior distribution with the actual empirical distribution $p(z)$ of a network with ReLU activations.

In a network implemented using Softplus activations, a log-normal is a good fit of the distribution of the activations. This is to be expected, especially when using batch-normalization, since the pre-activations will approximately follow a normal distribution with zero mean, and the Softplus approximately resembles a scaled exponential near zero. Therefore, in this case we suggest using a log-normal distribution as our prior $p(z)$. In Figure 1b, we compare this prior with the empirical distribution $p(z)$ of a network with Softplus activations.

Using these priors, we can finally compute the KL divergence term in eq. (2) for both ReLU activations and Softplus activations. We prove the following two propositions in Appendix A.

**Proposition 1** (Information dropout cost for ReLU activations). *Let $z = \varepsilon \cdot f(x)$, where $\varepsilon \sim p_\alpha(\varepsilon)$, and assume $p(z) = q\delta_0(z) + c/z$. Then, assuming $f(x) \neq 0$, we have*

$$\mathrm{KL}(p_\theta(z|x) \parallel p(z)) = -H(p_{\alpha(x)}(\log \varepsilon)) + \log c$$

*In particular, if $p_\alpha(\varepsilon)$ is chosen to be the log-normal distribution $p_\alpha(\varepsilon) = \log \mathcal{N}(0, \alpha_\theta^2(x))$, we have*

$$\mathrm{KL}(p_\theta(z|x) \parallel p(z)) = -\log \alpha_\theta(x) + const. \tag{3}$$

*If instead $f(x) = 0$, we have*

$$\mathrm{KL}(p_\theta(z|x) \parallel p(z)) = -\log q.$$

**Proposition 2** (Information dropout cost for Softplus activations). *Let $z = \varepsilon \cdot f(x)$, where $\varepsilon \sim p_\alpha(\varepsilon) = \log \mathcal{N}(0, \alpha_\theta^2(x))$, and assume $p_\theta(z) = \log \mathcal{N}(\mu, \sigma^2)$. Then, we have*

$$\mathrm{KL}(p_\theta(z|x) \parallel p(z)) = \frac{1}{2\sigma^2} \left(\alpha^2(x) + \mu^2\right) - \log \frac{\alpha(x)}{\sigma} - \frac{1}{2}. \tag{4}$$

Substituting the expression for the KL divergence in eq. (3) inside eq. (2), and ignoring for simplicity the special case $f(x) = 0$, we obtain the following loss function for ReLU activations

$$\mathcal{L} = \frac{1}{N} \sum_{i=1}^{N} \mathbb{E}_{\mathbf{z} \sim p_\theta(\mathbf{z}|\mathbf{x}_i)} [\log p(\mathbf{y}_i|\mathbf{z})] + \beta \log \alpha_\theta(\mathbf{x}_i), \tag{5}$$

and a similar expression for the Softplus. Notice that the first expectation can be approximated by sampling (in the experiments we use one single sample, as customary for dropout), and is just the average cross-entropy term that is typical in deep learning. The second term, which is new, penalizes the network for choosing a low variance for the noise, i.e. for letting more information pass through to the next layer. This loss can be optimized easily using stochastic gradient descent and the reparametrization trick of Kingma & Welling (2013) to back-propagate the gradient through the sampling operation.

## 5 VARIATIONAL AUTOENCODERS AND INFORMATION DROPOUT

In this section, we briefly outline a link between *variational autoencoders* (Kingma & Welling, 2013) and Information Dropout. A variational autoencoder (VAE) aims to reconstruct, given a training dataset $\mathcal{D} = \{\mathbf{x}_i\}$, a latent random variable $\mathbf{z}$ such that the observed data $\mathbf{x}$ can be thought as being generated by the, usually simpler, variable $\mathbf{z}$ through some unknown generative process $p_\theta(\mathbf{x}|\mathbf{z})$. In practice, this is done by minimizing the negative variational lower-bound to the marginal log-likelihood of the data

$$\mathcal{L}(\theta) = \frac{1}{N} \sum_{i=1}^{N} \mathbb{E}_{\mathbf{z} \sim p_\theta(\mathbf{z}|\mathbf{x}_i)} [-\log p_\theta(\mathbf{x}_i|\mathbf{z})] + \mathrm{KL}(p_\theta(\mathbf{z}|\mathbf{x}_i) \parallel p(\mathbf{z})),$$

which can be optimized easily using the SGVB method of Kingma & Welling (2013). Interestingly, when the task is reconstruction, that is when $\mathbf{y} = \mathbf{x}$, the IB loss function in eq. (2) reduces to

$$\mathcal{L}(\theta) = \frac{1}{N} \sum_{i=1}^{N} \mathbb{E}_{\mathbf{z} \sim p_\theta(\mathbf{z}|\mathbf{x}_i)} [-\log p_\theta(\mathbf{x}_i|\mathbf{z})] + \beta \, \mathrm{KL}(p_\theta(\mathbf{z}|\mathbf{x}_i) \parallel p(\mathbf{z})).$$

Therefore, by letting $\beta = 1$ in the previous expression, we obtain exactly the loss function of a variational autoencoder, that is, the representation $\mathbf{z}$ computed by the Information Dropout layer coincides with the latent variable $\mathbf{z}$ computed by the VAE. This is in part to be expected, since the objective of Information Dropout is to create a representation of the data that is minimal sufficient for the task of reconstruction, and the latent variables of a VAE can be thought as such a representation. The term $\beta$ in this case can be seen as managing the trade off between the fidelity of the reconstruction of the input from the representation (measured by the cross-entropy), against the compression factor (complexity) of the representation (measured by the KL-divergence).

## 6 EXPERIMENTS

We compare our method with the Dropout baseline on several standard benchmark datasets using different networks architecture, and highlight a few key properties of Information Dropout. All the models were implemented using TensorFlow (Abadi et al., 2015). As Kingma et al. (2015) also notice, letting the variance of the noise increase too much leads to poor local minima in the optimization process. To avoid this problem, we put the constraint $\alpha(x) < 0.7$, so that the maximum variance of the log-normal error distribution will be approximatively 1, the same as binary dropout when using a drop probability of 0.5. In all experiments we divide the KL-divergence term by the number of training samples, so that for $\beta = 1$ the scaling of the KL-divergence term in similar to the one used by Variational Dropout (see Section 2).

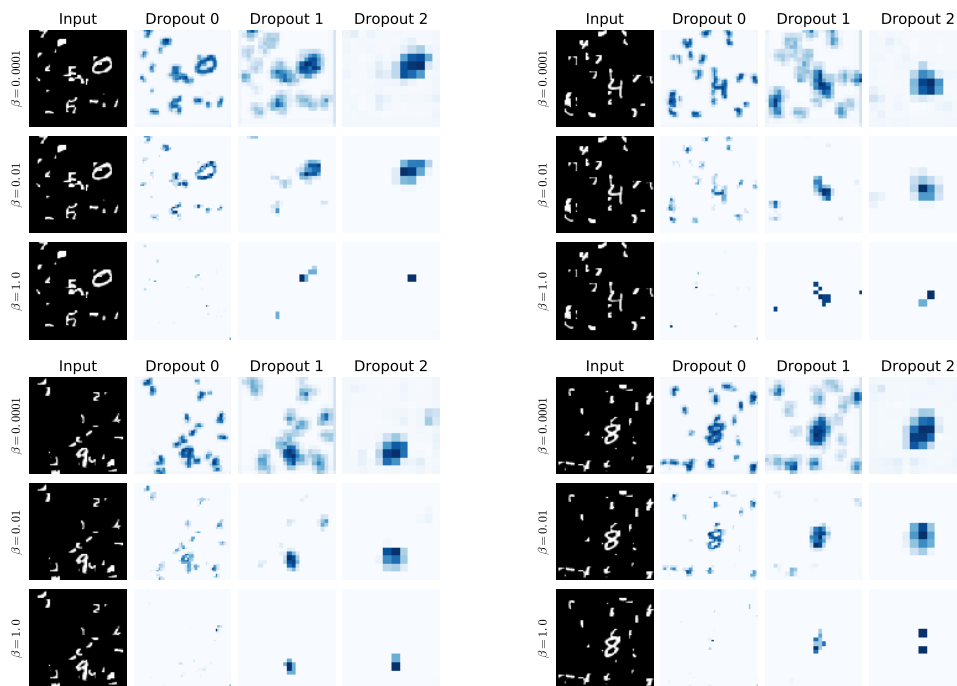

Figure 2: For four different input samples, we show the plot of the total KL-divergence at each spatial location in the first three Information Dropout layers of All-CNN-96 (see Table 2) trained on Cluttered MNIST with different values of $\beta$. This measures how much information from each part of the image the Information Dropout layer is letting flow to the next layer. While for low value $\beta$ information about the nuisances is still transmitted to the next layers, for higher value of $\beta$ the Information Dropout layers drop the information as soon as the receptive field is big enough to recognize it as a nuisance. The resulting representation is therefore more robust to nuisances, and provides better generalization performances. Unlike in classical dropout or Variational Dropout, the noise added by Information Dropout is tailored to the specific sample, to the point that the KL-divergence alone provides enough information to localize the digit.

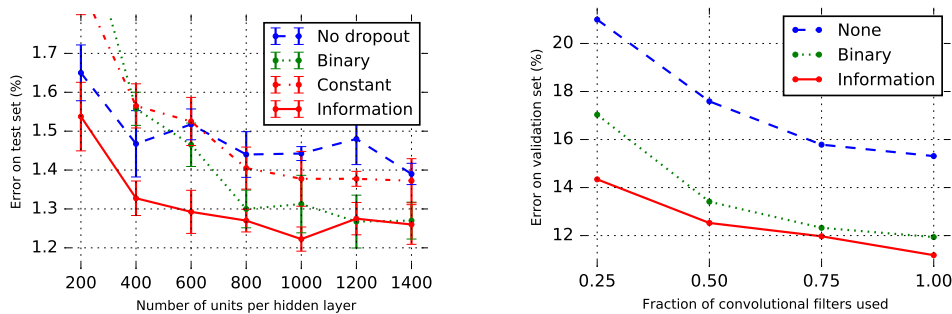

(a) Classification error on MNIST for different sizes  (b) Classification error on CIFAR-10 for different sizes

Figure 3: (a) Average classification error on MNIST over 3 runs of several dropout methods applied to a fully connected network with three hidden layers and ReLU activations. Information dropout outperforms binary dropout, especially on smaller networks, possibly because dropout severely reduces the already limited capacity of the network, while Information Dropout can adapt the amount of noise to the data and the size of the network. Information dropout also outperforms a dropout layer that uses constant log-normal noise with the same variance, confirming the benefits of adaptive noise. Variational dropout yields a similar performance to Information Dropout for a suitably chosen scaling factor, and is not shown in the plot. (b) Classification error on CIFAR-10 for several dropout methods applied to the All-CNN-32 network (see Table 2) using Softplus activations.

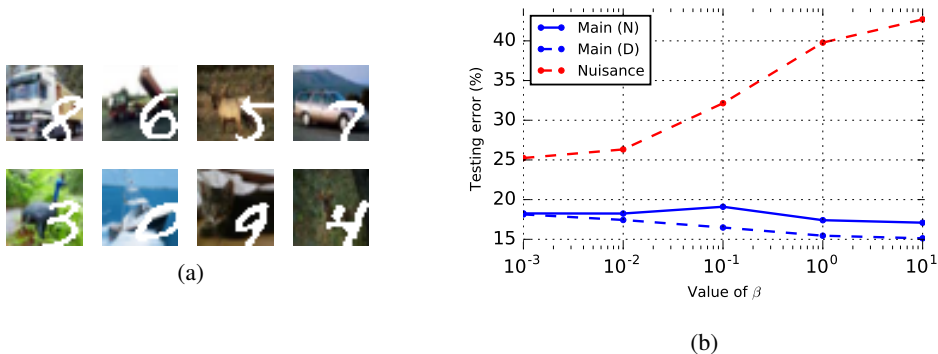

(a)

(b)

Figure 4: (a) A few samples from our Occluded CIFAR dataset. (b) Plot of the testing error on the main task (classifying the CIFAR image) and on the nuisance task (classifying the occluding MNIST digit) as $\beta$ varies. For both tasks, we use the same representation of the data trained for the main task using Information Dropout. For larger values of $\beta$ the representation is increasingly more invariant to nuisances, making the nuisance classification task harder, but improving the performance on the main task by preventing overfitting. For the nuisance task, we test using the learned noisy representation of the data, since we are interested specifically in the effects of the noise. For the main task, we show the result both using the noisy representation (N), and the deterministic representation (D) obtained by disabling the noise at testing time.

**Cluttered MNIST.** To visually asses the ability of Information Dropout to create a representation of the input that is increasingly insensitive to nuisance factors, we train the All-CNN-96 network (Table 2) for classification on a Cluttered MNIST dataset (cf. Mnih et al., 2014), consisting of 96x96 images containing a single MNIST digit together with 21 distraction objects. The dataset is divided in 50.000 training images and 10.000 testing images. As shown in Figure 2, for small values of $\beta$, the network lets through both the objects of interest (digits) and distractors, to upper layers. By increasing the value of $\beta$, we force the network to disregard something, and the network decides to disregard the least discriminative components of the data, thereby building a better representation for the task. This behavior depends on the ability of Information Dropout to learn the structure of the nuisances in the dataset which, unlike other methods, is facilitated by the ability to select noise level on a per-sample basis.

**Occluded CIFAR.** Occlusions are a fundamental type of nuisance in vision for which it is difficult to hand-design invariant representations. To assess that the approximate minimal sufficient representation produced by Information Dropout has this invariance property, we created a new dataset by occluding images from CIFAR-10 with digits from MNIST (Figure 4a). We train the All-CNN-32 network (Table 2) to classify the CIFAR image. The information relative to the occluding MNIST digit is then a nuisance for the task, and therefore should be excluded from the final representation. To test this, we train a secondary network to classify the nuisance MNIST digit using only the the representation learned for the main task. When training with small values of $\beta$, the network has very little pressure to limit the presence of nuisance information in the representation, so we expect the nuisance classifier to perform better. On the other hand, increasing the value of $\beta$ we expect the performance to degrade, since the representation will become increasingly minimal, and therefore invariant to nuisances. The results in Figure 4b confirm this intuition.

**MNIST and CIFAR-10.** Similar to Kingma et al. (2015), to see the effect of Information Dropout on different network sizes and architectures, we train on MNIST a network with 3 fully connected hidden layers with a variable number of hidden units, and we train on CIFAR-10 (Krizhevsky & Hinton, 2009) the All-CNN-32 convolutional network described in Table 2, using a variable percentage of all the filters. The fully connected network was trained for 80 epochs, using stochastic gradient descent with momentum with initial learning rate 0.07 and dropping the learning rate by 0.1 at 30 and 70 epochs. The CNN was trained for 160 epochs with initial learning rate 0.1 and dropping the learning rate by 0.1 at 80 and 120 epochs. We show the results in Figure 3. Information Dropout is comparable or outperforms binary dropout, especially on smaller networks. A possible explanation is

Table 2: Structure of the networks used in the experiments. The design of network is based on Springenberg et al. (2014), but we also add batch normalization before the activations of each layer. Depending on the experiment, the ReLU activations are replaced by Softplus activations, and the dropout layer is implemented with binary dropout, Information Dropout or completely removed.

(a) All-CNN-32

| Input 32x32 |
| --- |
| 3x3 conv 96 ReLU |
| 3x3 conv 96 ReLU |
| 3x3 conv 96 ReLU stride 2 |
| dropout |
| 3x3 conv 192 ReLU |
| 3x3 conv 192 ReLU |
| 3x3 conv 192 ReLU stride 2 |
| dropout |
| 3x3 conv 192 ReLU |
| 1x1 conv 192 ReLU |
| 1x1 conv 10 ReLU |
| spatial average |
| softmax |

(b) All-CNN-96

| Input 96x96 |
| --- |
| 3x3 conv 32 ReLU |
| 3x3 conv 32 ReLU |
| 3x3 conv 32 ReLU stride 2 |
| dropout |
| 3x3 conv 64 ReLU |
| 3x3 conv 64 ReLU |
| 3x3 conv 64 ReLU stride 2 |
| dropout |
| 3x3 conv 96 ReLU |
| 3x3 conv 96 ReLU |
| 3x3 conv 96 ReLU stride 2 |
| dropout |
| 3x3 conv 192 ReLU |
| 3x3 conv 192 ReLU |
| 3x3 conv 192 ReLU stride 2 |
| dropout |
| 3x3 conv 192 ReLU |
| 1x1 conv 192 ReLU |
| 1x1 conv 10 ReLU |
| spatial average |
| softmax |

that dropout severely reduces the already limited capacity of the network, while Information Dropout can adapt the amount of noise to the data and to the size of the network so that the relevant information can still flow to the successive layers.

**VAE.** To validate what we said in Section 5, we replicate the basic variational autoencoder of Kingma & Welling (2013), implementing it both with Gaussian latent variables, as in the original, and with an Information Dropout layer. We trained both implementations for 300 epochs dropping the learning rate by 0.1 at 30 and 120 epochs. We report the results in the following table. The Information Dropout implementation has similar performance to the original, confirming that a variational autoencoder can be considered a special case of Information Dropout.

Table 1: Average variational lower-bound $\mathcal{L}$ on the testing dataset for a simple VAE, where the size of the latent variable $\mathbf{z}$ is $256 \cdot k$ and the encoder/decoder each contain $512 \cdot k$ hidden units. The latent variable $\mathbf{z}$ is implemented either using a Gaussian vector or using Information Dropout. Both methods achieve a similar performance.

| $k$ | Gaussian | Information |
| --- | --- | --- |
| 1 | -98.75 | -100.04 |
| 2 | -99.03 | -99.07 |
| 3 | -98.72 | -99.10 |

## 7    CONCLUSION

We introduced a new dropout method that can be seen as an efficient implementation of the Information Bottleneck principle and that helps the network learn the structure of the nuisance factors affecting the data and build representations that are insensitive to those nuisances, therefore improving generalization performance. We also analyzed from an information theoretic viewpoint the role that noise injected in a network has in learning nuisance invariance. Experimental evidence confirms the insights stemming from the theory thus developed.

Another interpretation of Information Dropout is as a way of biasing the network towards reconstructing representations of the data that are compatible with a Markov chain generative model,

making it more suited to data coming from hierarchical models, and in this sense is complementary to architectural constraint, such as convolutions, that instead bias the model toward geometrical tasks.

An important topic in representation learning, which we did not discuss explicitly, is whether we can also learn a "disentangled" representation of the data, and whether the factors of this representation are related to the latent factors of the real generative model. In Appendix B and Appendix C, we show that by adding independent (multiplicative) noise to the activations and by using the IB loss, we naturally favor representations which have mutually independent components and, in some restricted situations, we prove that we can disentangle the relevant part of the information from the nuisance variability. We leave proving more general results in this direction to a future work.

ACKNOWLEDGMENTS

Work supported by ARO, ONR, AFOSR.

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

## A    COMPUTATION OF THE KL-DIVERGENCE

**Proposition** (Information dropout cost for ReLU activations)**.** *Let $z = \varepsilon \cdot f(x)$, where $\varepsilon \sim p_\alpha(\varepsilon)$, and assume $p(z) = q\delta_0(z) + c/z$. Then, assuming $f(x) \neq 0$, we have*

$$\mathrm{KL}(p_\theta(z|x) \parallel p(z)) = -H(p_{\alpha(x)}(\log \varepsilon)) + \log(c)$$

*In particular, if $p_\alpha(\varepsilon)$ is chosen to be the log-normal distribution $p_\alpha(\varepsilon) = \log \mathcal{N}(0, \alpha_\theta^2(x))$, we have*

$$\mathrm{KL}(p_\theta(z|x) \parallel p(z)) = -\log \alpha_\theta(x) + const.$$

*If instead $f(x) = 0$, we have*
$$\mathrm{KL}(p_\theta(z|x) \parallel p(z)) = -\log q.$$

*Proof.* If $f(x) \neq 0$, then we also have $z \neq 0$. Since the KL-divergence is invariant under parameter transformations we can write

$$\mathrm{KL}(p_\theta(z|x) \parallel p_\theta(z)) = \mathrm{KL}(p_\theta(\log z|x) \parallel p_\theta(\log z))$$
$$= \int \log \left( \frac{p_\theta(\log z|x)}{p_\theta(\log z)} \right) p_\theta(\log z|x) dz$$
$$= \int \log \left( p_{\alpha(x)}(\log \varepsilon) \right) p_{\alpha(x)}(\log \varepsilon) d\varepsilon - \log c$$
$$= -H(p_{\alpha(x)}(\log \varepsilon)) - \log c.$$

For the second part, notice that by definition $p_{\alpha(x)} = \mathcal{N}(0, \alpha_\theta^2(x))$ and

$$H(\mathcal{N}(0, \alpha)) = \log \alpha_\theta(x) + \frac{1}{2} \log(2\pi e).$$

Finally, if $f(x) = 0$, then also $z = 0$, so $p(z|x) = \delta_0(z)$. It is then easy to see that
$$\mathrm{KL}(p_\theta(z|x) \parallel p(z)) = -\log p(z = 0) = -\log q.$$

$\square$

**Proposition** (Information dropout cost for Softplus activations)**.** *Let $z = \varepsilon \cdot f(x)$, where $\varepsilon \sim p_\alpha(\varepsilon) = \log \mathcal{N}(0, \alpha_\theta^2(x))$, and assume $p_\theta(z) = \log \mathcal{N}(\mu, \sigma^2)$. Then, we have*

$$\mathrm{KL}(p_\theta(z|x) \parallel p(z)) = \frac{1}{2\sigma^2} \left( \alpha^2(x) + \mu^2 \right) - \log \frac{\alpha(x)}{\sigma} - \frac{1}{2}.$$

*Proof.* Since the KL divergence is invariant for reparametrizations, the divergence between two log-normal distributions is equal to the divergence between the corresponding normal distributions. Therefore, using the known formula for the KL divergence of normals, we get the desired result. $\square$

## B    EFFECTS OF USING A FACTORIZED APPROXIMATE PRIOR

In Information Dropout, we want to find a representation $\mathbf{z} \sim q(\mathbf{z}|\mathbf{x})$ that minimizes the objective

$$H_q(\mathbf{y}|\mathbf{z}) + \beta I(\mathbf{z}; \mathbf{x}).$$

This objective can be rewritten as

$$H_q(\mathbf{y}|\mathbf{z}) + \beta \mathbb{E}_x[\mathrm{KL}(q(\mathbf{z}|\mathbf{x}) \parallel q(\mathbf{z}))],$$

where the parameters of the posterior distribution $q(\mathbf{z}|\mathbf{x})$ are learned using a neural network. The prior distribution $q(\mathbf{z})$ should in theory be computed from the posterior by integrating over $\mathbf{x}$. However, since this is not feasible, we instead introduce a factorized approximation of the prior in the form $p(\mathbf{z}) = \prod_j p_j(z_j)$, and solve the problem

$$H_q(\mathbf{y}|\mathbf{z}) + \beta\mathbb{E}_x[\mathrm{KL}(q(\mathbf{z}|\mathbf{x}) \,\|\, p(\mathbf{z}))].$$

In this appendix, we show that the latter problem is indeed equivalent to the former when the components of the representation are mutually independent, and that assuming a factorized prior automatically favors the creation of such representations. In the following proposition, for simplicity, we concentrate on discrete random variables. Recall that the *total correlation*, or *multivariate mutual information*, of a variable $\mathbf{z} = (z_1, \dots, z_n)$ is defined as

$$\mathrm{TC}(\mathbf{z}) = \sum_j H(z_j) - H(\mathbf{z})$$

$$= \mathrm{KL}(q(\mathbf{z}) \,\|\, \textstyle\prod_j q_j(z_j)),$$

and that $\mathrm{TC}(\mathbf{z}) = 0$ if and only if the components of $\mathbf{z}$ are mutually independent.

**Proposition 3.** *Let $\mathbf{z} = (z_1, \dots, z_n)$ be a discrete random variable, let $q(\mathbf{z}|\mathbf{x})$ be a generic probability distribution, and let $p(\mathbf{z}) = \prod_{i=1^n} p(z_i)$ be a factorized prior distribution. Then, for any function $F(q)$, a minimization problem in the form*

$$\text{minimize}_{q,p} \quad F(q) + \beta\mathbb{E}_x[\mathrm{KL}(q(\mathbf{z}|\mathbf{x}) \,\|\, p(\mathbf{z}))],$$

*is equivalent to*

$$\text{minimize}_q \quad F(q) + \beta\left\{ I_q(\mathbf{z}; \mathbf{x}) + \mathrm{TC}_q(\mathbf{z}) \right\},$$

*where $I_q(\mathbf{z}; \mathbf{x})$ is the mutual information and $\mathrm{TC}_q(\mathbf{z})$ is the total correlation of $\mathbf{z}$, assuming $\mathbf{z} \sim q(\mathbf{z})$.*

*Proof.* To prove the proposition, we just need to minimize with respect to $p$ and substitute back the solution. Adding a Lagrange multiplier for the constrain $\sum_{z_i} p_i(z_i) = 1$, the problem can be rewritten as

$$\mathcal{L}(q, p) = F(q) + \beta\mathbb{E}_x\left[\sum_{\mathbf{z}} q(\mathbf{z}|\mathbf{x}) \log \frac{q(\mathbf{z}|\mathbf{x})}{p(\mathbf{z})} d\mathbf{z}\right] + \lambda\left(\sum_{z_i} p_i(z_i) - 1\right)$$

$$= F(q) + \beta\mathbb{E}_x\left[\sum_{\mathbf{z}} q(\mathbf{z}|\mathbf{x}) \log \frac{q(\mathbf{z}|\mathbf{x})}{\prod_{j=1}^n p_j(z_j)} d\mathbf{z}\right] + \lambda\left(\sum_{z_i} p_i(z_i) - 1\right).$$

Taking the derivative with respect to to $p_i(\bar{z}_i)$ we have

$$\frac{\partial\mathcal{L}(q, p)}{\partial p_i(\bar{z}_i)} = \beta\mathbb{E}_x\left[\sum_{\mathbf{z}} q(\mathbf{z}|\mathbf{x}) \log \frac{q(\mathbf{z}|\mathbf{x})}{\prod_{j=1}^n p_j(z_j)}\right] + \lambda$$

$$= -\beta \sum_{z_i = \bar{z}_i} \frac{\mathbb{E}_x[q(\mathbf{z}|\mathbf{x})]}{p_i(\bar{z}_i)} + \lambda$$

$$= -\beta \frac{q(\bar{z}_i)}{p(\bar{z}_i)} + \lambda.$$

Setting it to zero, we obtain $p(z_i) = q(z_i)$, that is, the optimal factorized prior is the product of the marginals. Substituting it back in the second term (the only one containing $p$), we obtain

$$\mathbb{E}_x[\mathrm{KL}(q(\mathbf{z}|\mathbf{x}) \,\|\, p(\mathbf{z}))] = \mathbb{E}_x\left[\sum_{\mathbf{z}} q(\mathbf{z}|\mathbf{x}) \log \frac{q(\mathbf{z}|\mathbf{x})}{\prod_{j=1}^n q_j(z_j)}\right]$$

$$= \mathbb{E}_x\left[\sum_{\mathbf{z}} q(\mathbf{z}|\mathbf{x}) \left(\log \frac{q(\mathbf{z}|\mathbf{x})}{q(\mathbf{z})} + \log \frac{q(\mathbf{z})}{\prod_{j=1}^n q_j(z_j)}\right)\right]$$

$$= \mathbb{E}_x\left[\sum_{\mathbf{z}} q(\mathbf{z}|\mathbf{x}) \log \frac{q(\mathbf{z}|\mathbf{x})}{q(\mathbf{z})}\right] + \sum_{\mathbf{z}} \mathbb{E}_{\mathbf{x}}[q(\mathbf{z}|\mathbf{x})] \log \frac{q(\mathbf{z})}{\prod_{j=1}^n q_j(z_j)}$$

$$= I_q(\mathbf{z}; \mathbf{x}) + \mathrm{KL}(q(\mathbf{z}) \,\|\, \textstyle\prod_j q_j(z_j))$$

$$= I_q(\mathbf{z}; \mathbf{x}) + \mathrm{TC}_q(\mathbf{z}).$$

$\square$

## C    INFORMATION BOTTLENECK PRINCIPLE AND DISENTANGLEMENT

In Appendix B, we showed that the loss function we use, in conjunction with a factorized prior, automatically favors representations of the data in which all the components are independent. In this appendix we show that, in the simple case of the data $\mathbf{x}$ coming from a Gaussian distribution, where our task is to recover one component of the data from a noisy observation, the solution of the Information Bottleneck Lagrangian when noise is added to the computation process naturally leads to recovering the independent components of the generative model. For simplicity, here we will use additive noise from a Gaussian distribution, but it is easily seen by exponentiating each random variable that this is exactly equivalent to using multiplicative noise from a log-normal distribution, which is the same case treated in the main paper.

Suppose we have a 2-dimensional Gaussian random variable

$$\mathbf{x} = (x, y) \sim \mathcal{N}(\mathbf{0}, I),$$

and that our task is to recover $y = \mathbf{e}_2^T \mathbf{x}$, given a noisy observation

$$\hat{\mathbf{x}} = A\mathbf{x} + \xi,$$

where $A$ is an orthogonal matrix and $\xi \sim \mathcal{N}(\mathbf{0}, \sigma_\xi^2)$ is some additive noise. We want to find an optimal representation of the input $\hat{\mathbf{x}}$ for the task $y$. To simplify the problem, we restrict to the representations in the form

$$\mathbf{z} = B\hat{\mathbf{x}},$$

where $B$ is an orthogonal matrix. Intuitively, the best representation for the task should be obtained by choosing $B = A^{-1}$, since then we would have $\mathbf{z} = A^{-1}\hat{\mathbf{x}} = \mathbf{x} + A^{-1}\xi$, and the component of $\hat{\mathbf{x}}$ relevant to the task, that is $y$, would be disentangled from the nuisance $x$. However, as we show in the following proposition, if we evaluate the representation using the conditional cross-entropy loss $\mathcal{L} = H(y|\mathbf{z})$, then all the choices of $B$ are equivalent, while if we add noise to the representation and use the IB loss, we naturally obtain that the optimal representation is disentangled.

**Proposition 4.** *Let $\mathbf{x} \sim \mathcal{N}(\mathbf{0}, I)$, $\hat{\mathbf{x}} = A\mathbf{x} + \xi$, where $A$ is orthogonal and $\xi \sim \mathcal{N}(\mathbf{0}, \sigma_\xi^2)$. We want to find a representation $\mathbf{z}$ of $\mathbf{x}$ for the task $y = \mathbf{e}_2^T \mathbf{x}$.*

*(i) Assume the representation $\mathbf{z}$ is in the form $\mathbf{z} = B\hat{\mathbf{x}}$, where $B$ is an orthogonal matrix. Then the cross-entropy loss $\mathcal{L} = H(y|\mathbf{z})$ does not depend on choice of $B$, so all the representations are equivalent for the cross-entropy loss.*

*(ii) Assume the representation $\mathbf{z}$ is in the form $\mathbf{z} = B\hat{\mathbf{x}} + \epsilon$, where $B$ is an orthogonal matrix and the noise $\epsilon$ has distribution $\epsilon \sim \mathcal{N}(\mathbf{0}, \Sigma = \mathrm{diag}(\sigma_1^2, \sigma_2^2))$. Without loss of generality, assume that $\sigma_2 \leq \sigma_1$. Then, for $\beta$ small enough, the representation that minimizes the IB Lagrangian*

$$\mathcal{L} = H(y|\mathbf{z}) + \beta I(\hat{\mathbf{x}}, \mathbf{z}),$$

*is the disentangled representation obtained by choosing $B = A^{-1}$ and, for $\beta \to 0^+$, we have $\sigma_1 \to +\infty$ and $\sigma_2 \to 0$, so the added noise selectively preserve only information relative to the task.*

*Proof.* For (i), notice that, since $y = \mathbf{e}_2^T \mathbf{x} = \mathbf{e}_2^T (C^{-1}\mathbf{z} - B\xi)$ we have

$$y|\mathbf{z} \sim \mathcal{N}(\mathbf{e}_2^T C^{-1}\mathbf{z}, \sigma_\xi^2),$$

therefore,

$$H(y|\mathbf{z}) = \log \sigma_\xi + \frac{1}{2}\log(2\pi e)$$

is independent from $B$.

For (ii), reasoning as before, we have

$$y|\mathbf{z} \sim \mathcal{N}(\mathbf{e}_2^T C^{-1}\mathbf{z}, \sigma_\xi^2 + \mathbf{e}_2^T C^T \Sigma C \mathbf{e}_2),$$

where we have defined $C = BA$. Since $C$ is orthogonal, we can write $C\mathbf{e}_2 = (\sin\theta, \cos\theta)$ for some $\theta$. Notice in particular that for $\theta = 0$ we would have $C = I$, and so $B = A^{-1}$. Using this, we can rewrite the expression above as

$$y|\mathbf{z} \sim \mathcal{N}(\mathbf{e}_2^T C^{-1}\mathbf{z}, \sigma_\xi^2 + \sigma_1^2 \sin^2\theta + \sigma_2^2 \cos^2\theta).$$

Therefore, the conditional entropy is

$$H(y|\mathbf{z}) = \log(\sigma_\xi^2 + \sigma_1^2 \sin^2 \theta + \sigma_2^2 \cos^2 \theta) + const.$$

We now need to compute the mutual information $I(\hat{\mathbf{x}}; \mathbf{z})$ term of the IB Lagrangian. Recall that the mutual information between two N-dimensional Gaussian variables $\mathbf{x}$ and $\mathbf{z}$ is

$$I(\mathbf{x}; \mathbf{z}) = \frac{1}{2} \log \left( \frac{|\Sigma_{\mathbf{xx}}||\Sigma_{\mathbf{zz}}|}{|\Sigma|} \right),$$

where $\Sigma_{\mathbf{xx}}$ and $\Sigma_{\mathbf{zz}}$ are the covariances matrixes of $\mathbf{x}$ and $\mathbf{z}$ respectively and $\Sigma$ is the covariance matrix of the pair $(\mathbf{x}, \mathbf{z})$. Using this, we obtain

$$I(\hat{\mathbf{x}}, \mathbf{z}) = \frac{1}{2} \left[ \log \left( 1 + \frac{1 + \sigma_\xi^2}{\sigma_1^2} \right) + \log \left( 1 + \frac{1 + \sigma_\xi^2}{\sigma_2^2} \right) \right].$$

Putting everything together, the IB loss function is

$$\mathcal{L} = \log(\sigma_\xi^2 + \sigma_1^2 \cos^2 \theta + \sigma_2^2 \sin^2 \theta) + \frac{\beta}{2} \left[ \log \left( 1 + \frac{1 + \sigma_\xi^2}{\sigma_1^2} \right) + \log \left( 1 + \frac{1 + \sigma_\xi^2}{\sigma_2^2} \right) \right].$$

Since $\theta$ appears only in the first term, we can immediately minimize the loss with respect to $\theta$. Recall that we are assuming $\sigma_2 \leq \sigma_1$. Then, there are two possibilities: if $\sigma_2 < \sigma_1$, then we must have $\theta = 0$ (or equivalently $C = I$ and hence $B = A^{-1}$). If instead $\sigma_1 = \sigma_2$, then all $\theta$ are equivalent. In both cases, the loss function simplifies to

$$\mathcal{L} = \log(\sigma_\xi^2 + \sigma_2^2) + \frac{\beta}{2} \left[ \log \left( 1 + \frac{1 + \sigma_\xi^2}{\sigma_1^2} \right) + \log \left( 1 + \frac{1 + \sigma_\xi^2}{\sigma_2^2} \right) \right].$$

The terms containing $\sigma_1$ are now separate from those containing $\sigma_2$ and we can minimize them independently. Doing so we obtain

$$\sigma_1 \to +\infty,$$
$$\sigma_2 = f(\beta)$$

where $f(\beta)$ is a function of $\beta$ such that $f(\beta) \to 0$ as $\beta \to 0^+$. Finally, since we have now established that for $\beta$ small enough we have $\sigma_2 < \sigma_1$, we can conclude from what we said before that we must have $\theta = 0$, and therefore that $B = A^{-1}$ and the representation is disentangled, as we wanted. $\square$

While the previous proposition deals with a very simple case, under restrictive hypotheses on the form of the representation, we conjecture that a similar property should hold more generally for any representation. For example, assume that we can find a mapping $f(\mathbf{x}) = (T(\mathbf{x}), n)$, where $T(\mathbf{x})$ is a sufficient statistic of the data for the task $\mathbf{y}$, while $n$ is independent from $y$. That is, $f(\mathbf{x})$ can "disentangle" the component of the data relevant to the task from the nuisances. Then, it would be easy to minimize the IB Lagrangian by routing all the noise on $n$, while leaving $T(\mathbf{x})$ untouched. We claim that the opposite should also happen: When the IB Lagrangian is minimized, the representation should be decomposed in a part relevant for the task, and a part which is independent. We leave further study of this problem to a future work.

# D  ADDITIONAL PLOTS

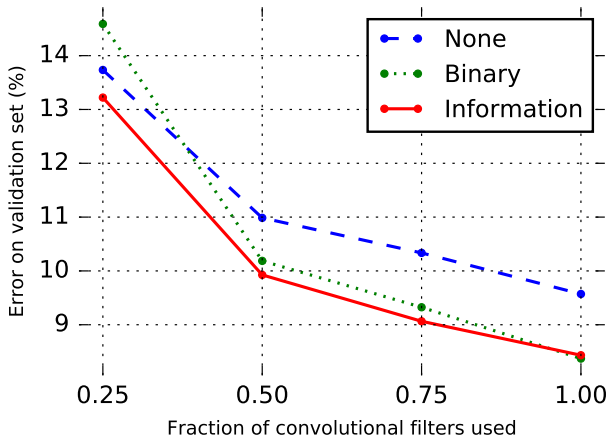

Figure 5: Classification error on CIFAR-10 for several dropout methods applied to the All-CNN-32 network (Table 2) using ReLU activations and varying the number of filters used. While in Figure 3b we used controlled setting to provide a fairer comparison, here we use the same settings suggested by Springenberg et al. (2014). In particular we add a weight decay factor of 0.001, reduce the batch size to 128, and drop the inputs with probability 0.2. The main effect of Information Dropout is to dynamically reduce the information flow in the network. Since the same can be achieved by carefully tuning the dropout rate and/or the number of filters used, we expect binary dropout to performs similarly on a finely tuned standard architecture, as is the case here when using all the filters.

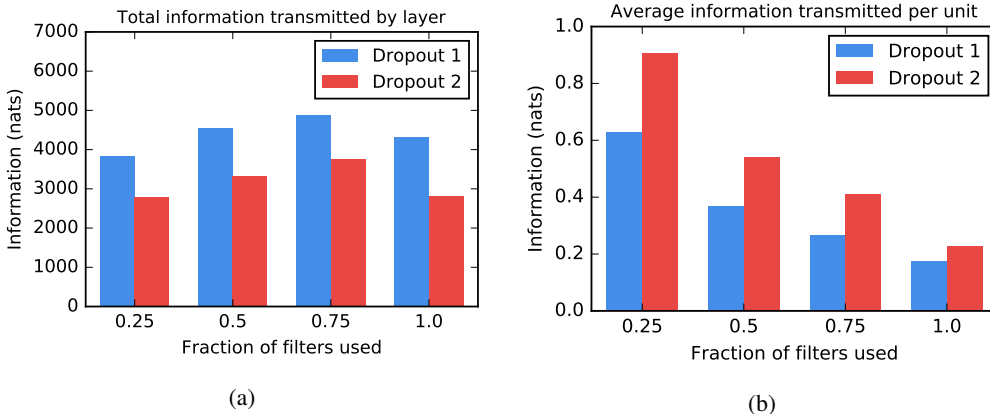

(a)                                                    (b)

Figure 6: Plots of (a) the total information transmitted through the two dropout layers of a All-CNN-32 network with Softplus activations trained on CIFAR and (b) the average quantity of information transmitted through each unit in the two layers. From (a) we see that the total quantity of information transmitted does not vary much with the number of filters and that, as expected, the second layer transmits less information than the first layer, since prior to it more nuisances have been disentangled and discarded. In (b) we see that when we decrease the number of filters, we force each single unit to let more information flow (i.e. we apply less noise), and that the units in the top dropout layer contain on average more information relevant to the task than the units in the bottom dropout layer.

