# Peer review of "Information Dropout: learning optimal representations through noise"

_ICLR 2017 — rejected_

[Author Response · Alessandro Achille · 07 Dec 2016]
**Paper update**

A personal communication asked whether there are cases in which a stochastic representation of the data can obtain a better value of the IB Lagrangian than any deterministic representation; in response to this, we added a remark in Section 3 saying that this indeed can happen. 

In response to a question by the reviewer, we added to Section 2 a few examples of nuisances that act as a group on the data.

We updated the MNIST and CIFAR experiments: all the qualitative results are the same as before, but we slightly changed the hyperparameters and the optimization method to provide a more accurate and fairer comparison between the algorithms.

Finally, we added an appendix to fill a gap in the narrative between Equation (2),  where the two distributions in the KL term were the actual prior and posterior of z, and Section 4, where we assume an approximated prior whose parameters are learned independently. Specifically, we show that if the approximated prior of the activations is chosen to be factorized, as we do, then our loss function differs from the actual IB Lagrangian by the total correlation of z. As a consequence, our approximation is correct when the components of z are mutually independent, and the loss function we use actually encourages this independence.

We would like to thank all the people that gave us early feedback on the paper.

[Official Review · AnonReviewer1 · rating 6 · confidence 4 · 16 Dec 2016]
**Interesting theory, but experimental results not (yet) very convincing, unfortunately.**

An interesting connection is made between dropout, Tishby et al's "information bottleneck" and VAEs. Specifically, classification of 'y' from 'x' is split in two faces: an inference model z ~ q(z|x), a prior p(z), and a classifier y ~ p(y|z). By optimizing the objective E_{(x,y)~data} [ E_{z~q(z|x)}[log p(x|y)] + lambda * KL(q(z|x)||p(z))], with lambda <= 1, an information bottleneck 'z' is formed, where lambda controls an upper bound on the number of bits traveling through 'z'.

The objective is equivalent to a VAE objective with downweighted KL(posterior|prior), an encoder that takes as input 'x', and a decoder that only predicts 'x'.

- Related work (section 2) is discussed sufficiently. 
- In section 3, would be better to remind us the definition of mutual information.
- Connection to VAEs in section 5 is interesting.
- Unfortunately, the MNIST/CIFAR-10 results are not great. Since the method is potentially more flexible than other forms of dropout, this is slightly disappointing.
- It's unclear why the CIFAR-10 results seem to be substantially worse than the results originally reported for that architecture.
- It's unclear which version of 'beta' was used in figure 3a.

Overall I think the theory presented in the paper is promising. However, the paper lacks sufficiently convincing experimental results, and I encourage the authors to do further experiments that prove significant improvements, at least on CIFAR-10, perhaps on larger problems.

[Official Review · AnonReviewer3 · rating 4 · confidence 4 · 16 Dec 2016 (modified: 19 Jan 2017)]
**interesting idea, but not very convincing**

The authors propose "information dropout", a variation of dropout with an information theoretic interpretation. A dropout layer limits the amount of information that can be passed through it, and the authors quantify this using a variational bound. 

It remains unclear why such an information bottleneck is a good idea from a theoretical standpoint. Bayesian interpretations lend a theoretical basis to parameter noise, but activation noise has no such motivation. The information bottleneck indeed limits the information that can be passed through, but there is no rigorous argument for why this should improve generalization.

The experiments are not convincing. The CIFAR-10 results are worse than those in the paper that originally proposed the network architecture they use (Springenberg et al). The VAE results on MNIST are also horrible.

[Official Review · AnonReviewer2 · rating 6 · confidence 4 · 19 Dec 2016]
**Insightful theoretical derivation, experiments can be improved.**

Paper summary
This paper develops a generalization of dropout using information theoretic
principles. The basic idea is that when learning a representation z of input x
with the aim of predicting y, we must choose a z such that it carries the least
amount of information about x, as long as it can predict y. This idea can be
formalized using the Information Bottleneck Lagrangian. This leads to an
optimization problem which is similar to the one derived for variational
dropout, the difference being that Information dropout allows for a scaling
factor associated with the KL divergence term that encourages noise. The amount
of noise being added is made a parameterized function of the data and this
function is optimized along with the rest of the model. Experimental results on
CIFAR-10 and MNIST show (small) improvements over binary dropout.

Strengths
- The paper highlights an important conceptual link between probabilistic
  variational methods and information theoretic methods, showing that dropout
can be generalized using both formalisms to arrive at very similar models.
- The presentation of the model is excellent.
- The experimental results on cluttered MNIST are impressive.

Weaknesses
- The results on CIFAR-10 in Figure 3(b) seem to be on a validation set (unless
  the axis label is a typo). It is not clear why the test set was not used. This
makes it hard to compare to results reported in Springenberg et al, as well as
other results in literature.

Quality
The theoretical exposition is high quality. Figure 2 gives a nice qualitative
assessment of what the model is doing. However, the experimental results
section can be made better, for example, by matching the results on CIFAR-10 as
reported in Springenberg et al. and trying to improve on those using information
dropout.

Clarity
The paper is well written and easy to follow.

Originality
The derivation of the information dropout optimization problem using IB
Lagrangian is novel. However, the final model is quite close to variational
dropout.

Significance
This paper will be of general interest to researchers in representation learning
because it highlights an alternative way to think about latent variables (as
information bottlenecks). However, unless the model can be shown to achieve
significant improvements over simple dropout, its wider impact is likely to be
limited.

Overall
The paper presents an insightful theoretical derivation and good preliminary
results. The experimental section can be improved.

Minor comments and suggestions -
- expecially -> especially
- trough -> through
- There is probably a minus sign missing in the expression for H(y|z) above Eq (2).
- Figure 3(a) has error bars, but 3(b) doesn't. It might be a good idea to have those
for Figure 3(b) as well.
- Please consider comparing Figure 2 with the activity map of a standard CNN
  trained with binary dropout, so we can see if similar filtering out is
happening there already.

[Author Response · Alessandro Achille · 03 Jan 2017]
**More details on experiments and further tests**

We thank all the reviewers for their comments. We would like to provide some clarification regarding the experiments in the paper, and address some of the concerns which were raised.

>> The CIFAR-10 results are worse than those in the paper that originally proposed the network architecture they use (Springenberg et al). The VAE results on MNIST are also horrible.

If we use exactly the same architecture of Springenberg et al., then our results on CIFAR are, as predicted by the theory, comparable asymptotically, and better for smaller nets. We have added experiments that show this in the revised version to be uploaded soon. Also, our results on VAE are comparable to [KW13] for a similar architecture.

Note, however, that the goal of our experiments is not to improve state-of-the-art on CIFAR-10 or MNIST, but to illustrate the effect of Information Dropout when compared to other forms of dropout, and to validate the intuition derived from the theory. For this reason, for the experiments in the paper we chose the simplest empirical settings, and modified the All Convolutional Net to isolate potentially confounding factors: we removed weight decay, increased the batch size to reduce gradient noise, simplified the architecture by removing the initial dropout layer, and used less aggressive learning rates and no fine tuning.  We also replaced ReLU with Softplus to make the results comparable with those of [KSW15]. This also served to validate the theory which applies to both ReLU and Softplus. 

Many factors affect empirical performance, only few of which are relevant to validating our theory. To the latter hand, we went to great length to ensure that the experiments are *controlled*. Only under careful control can the experiments be convincing in validating the theory.

Nevertheless, as suggested by the reviewers, we are currently exploring other experiments that would further illustrate the tradeoff between invariance to nuisances and sufficiency as mediated by the coefficient \beta. We will add these along with the further tests using the same architecture of Springerberg, as described above.

>> The results on CIFAR-10 in Figure 3(b) seem to be on a validation set

We are using the same nomenclature of [KSW15], since we want to make a direct comparison with their experiment. As customary for CIFAR, the data is divided into a disjoint training set (50,000 samples) and validation/test set (10,000 samples). We feel that "validation" here is more appropriate.

[KSW15] Diederik Kingma, Tim Salimans, and Max Welling, "Variational Dropout and the Local Reparameterization Trick", 2015

[KW13] Diederik P Kingma, Max Welling, "Auto-Encoding Variational Bayes", 2013

[Author Response · Alessandro Achille · 10 Jan 2017]
**Paper update: new experiments**

Following the suggestions of the reviewers, we updated the paper with new experiments and plots.

First, to empirically validate that, by increasing the value of the parameter \beta, we can obtain representations that are increasingly minimal and invariant while remaining discriminative (sufficient), we created a new dataset, called ‘Occluded CIFAR’. The experiment in Sec. 6 shows precisely this effect, thus validating the theoretical intuition. Indeed, by increasing \beta, we also prevent overfitting and the overall quality of the representation actually improves.

In Figure 5 in Appendix D, we added a comparison between Information Dropout and binary dropout using the same settings as [Springenberg et al., 2014]. For both methods we obtain a slightly better testing error than the original paper and, as also observed in the previous experiments, Information Dropout performs comparably or better than dropout.

In Figure 6, we plot the amount of information flowing through the dropout layers of a CNN as the number of filters varies. This plots supports some of the theoretical intuitions, and we show empirically that information dropout automatically selects a lower noise level for smaller networks, and that the units in the higher layers contain on average more information relative to the task than units in the bottom layers.

[Final Decision · Program Chairs · 06 Feb 2017]
**ICLR committee final decision**

The authors all agree that the theory presented in the paper is of high quality and is promising but the experiments are not compelling. The reviewers are concerned that the presented idea and connections to existing methods, while neat, may not be impactful as the promise of the theory does not bear out in practice. One reviewer is concerned that the presented theory is still not useful, stating that the "information bottleneck thus only becomes meaningful when the capacity of the encoding network is controlled in some measurable way, which is not discussed in the paper". In general, they seem to agree that the experimental evaluation is still preliminary and unfinished. As such, it would seem that the authors could make the paper far more compelling by demonstrating more compelling improvements on benchmark experiments and submitting to a future conference.